# How Do Road Traffic Noise and Residential Greenness Correlate with Noise Annoyance and Long-Term Stress? Protocol and Pilot Study for a Large Field Survey with a Cross-Sectional Design

**DOI:** 10.3390/ijerph20043203

**Published:** 2023-02-11

**Authors:** Javier Dopico, Beat Schäffer, Mark Brink, Martin Röösli, Danielle Vienneau, Tina Maria Binz, Silvia Tobias, Nicole Bauer, Jean Marc Wunderli

**Affiliations:** 1Empa, Swiss Federal Laboratories for Materials Science and Technology, 8600 Dubendorf, Switzerland; 2Federal Office for the Environment (FOEN), 3003 Bern, Switzerland; 3Swiss Tropical and Public Health Institute (Swiss-TPH), 4123 Allschwil, Switzerland; 4Faculty of Science, University of Basel, 4001 Basel, Switzerland; 5Institute of Forensic Medicine, University of Zurich (UZH), 8006 Zurich, Switzerland; 6Swiss Federal Institute for Forest, Snow and Landscape Research (WSL), 8903 Birmensdorf, Switzerland

**Keywords:** noise annoyance, road traffic noise, stress, green spaces, GIS

## Abstract

Urban areas are continuously growing, and densification is a frequent strategy to limit urban expansion. This generally entails a loss of green spaces (GSs) and an increase in noise pollution, which has negative effects on health. Within the research project RESTORE (Restorative potential of green spaces in noise-polluted environments), an extended cross-sectional field study in the city of Zurich, Switzerland, is conducted. The aim is to assess the relationship between noise annoyance and stress (self-perceived and physiological) as well as their association with road traffic noise and GSs. A representative stratified sample of participants from more than 5000 inhabitants will be contacted to complete an online survey. In addition to the self-reported stress identified by the questionnaire, hair cortisol and cortisone probes from a subsample of participants will be obtained to determine physiological stress. Participants are selected according to their dwelling location using a spatial analysis to determine exposure to different road traffic noise levels and access to GSs. Further, characteristics of individuals as well as acoustical and non-acoustical attributes of GSs are accounted for. This paper presents the study protocol and reports the first results of a pilot study to test the feasibility of the protocol.

## 1. Introduction

Urban areas with high-population density experience fast and continuous growth. Due to infill development (densification) as a frequent strategy to limit urban expansion, this is often accompanied by a qualitative and quantitative decline of green spaces (GSs). Moreover, urban areas are characterized by permanent noise-related activities, particularly by road traffic. Road traffic sound is broadband with dominant frequencies from 250 Hz to 2 kHz and covers a wide range of temporal patterns from intermittent to continuous [1]. Transportation noise, in particular from road traffic [2,3], is a major environmental problem, that primarily affects people in urban and semi-urban areas. In Europe, more than 125 million people are exposed to harmful environmental noise exceeding 55 dB *L*_den_ (A-weighted day-evening-night level) [2]. In Switzerland, more than one million inhabitants are affected by levels of traffic noise exceeding the noise exposure limit values of the Swiss legislation, 90% of whom live in and around large agglomerations [3]. Approximately 70% of the world’s population is expected to live in cities by 2050 [4], which will further aggravate the problem. As a consequence, high levels of activity—and thereby high noise levels—are likely to lead to increasingly elevated levels of stress which is detrimental to health [5,6]. The repercussions of noise effects on human health, wellbeing and cognitive performance have been highlighted in a broad range of studies (e.g., [2,7]). Noise annoyance and sleep disturbance are the most widespread effects across the population, and—as mediators—may trigger stress-related diseases [6]. For instance, in addition to preventing people from participating in leisure-time activities [8], long-term exposure to high noise levels may also contribute to obesity and diabetes [9,10]. Additionally, noise may also influence high blood pressure [6], trigger cardiovascular diseases [11] and even, affect mental health, resulting in depression and anxiety disorders [12,13].

Annoyance reactions are not fully explained by noise exposure levels alone. Annoyance is also linked with other aspects, including personal characteristics such as noise sensitivity [14], and the physical attributes of the environment that can influence the relationship between noise exposure with annoyance (e.g., [15]). A conclusive picture of the mechanisms of stress evolution due to environmental noise is not yet apparent in the literature; in particular, questions remain regarding the link between noise annoyance (as a psychological mediator) and stress (as a psychological and physiological outcome). Thus, more knowledge about the effect chain of noise annoyance, perceived stress and physiological stress levels is needed. Additionally, the potential effect of stress reduction by GSs is not well explored. There are strong indications from the literature that environmental resources such as access to a quiet place in the neighborhood and dwelling-related greenery may reduce stress. For instance, GSs were previously shown to reduce exposure to environmental stressors, help psychophysiological stress recovery, encourage physical activity and even reduce mortality [16,17]. Several studies have shown reductions in noise annoyance in relation to GSs [18,19,20]. Determination of which GS-related factors are crucial, e.g., perception or physical activity, is however not yet well understood.

Within the research project RESTORE, we aim to shed light on the relationships between noise, GSs, annoyance and stress. In this paper, the study protocol is presented, along with results from a pilot study that was conducted to test the feasibility of the study protocol.

## 2. Materials and Methods

### 2.1. Study Purpose, Research Hypotheses and Study Design

The purpose of the research project RESTORE (Restorative potential of green spaces in noise-polluted environments) is to study the effects of noise as an environmental stressor and impediment to recovering from stress and of GSs as a moderator. A further aim is to identify prerequisites of GSs to promote the reduction in noise-induced stress. The study protocol described here focuses on noise annoyance and long-term stress, provoked by road traffic noise exposure as the major technical environmental noise source, in a large field survey in the city of Zurich, Switzerland. We aim to assess the restorative potential of GSs in noise-polluted environments, to obtain new insights into the pathway from noise annoyance to physiological stress.

Our hypotheses are that (i) noise annoyance of people exposed to road traffic noise at home is associated with public GSs, (ii) self-reported stress of people exposed to road traffic noise at home is associated with public GSs and (iii) physiological stress of people exposed to road traffic noise at home is associated with public GSs.

To test our hypotheses, a cross-sectional study is carried out. The participants are selected and stratified according to both their noise exposure at home and access to GSs (see Section 2.2), with a focus on those with increased levels of road traffic noise. To identify the study sites in a stratified sample, the characteristics of interest are quantified in an explorative spatial analysis. The study consists of a field survey within a representative stratified sample of individuals, followed by a visit to the home of a subsample of the participants. The focus will be on noise annoyance; however, self-reported sleep disturbance will be examined as an additional outcome in all participants.

### 2.2. Spatial Analysis

For this study, GSs were defined as recreational areas with vegetation in a public, open and publicly accessible space. These include traditional urban parks and other spaces, such as urban forestry, public gardens, pocket parks or cemeteries, and may include built environmental features.

Exposure to road traffic noise during daytime (*L*_day_; see Section 2.3) and access to public GSs were spatially analyzed in the Geographic Information System (GIS) Esri ArcGIS (version 10.8.1).

Access to public GSs was derived from circular Euclidean buffers with a radius of 300 m around the buildings of the study participants (similar to [17,19]). Public GSs as identified by land-use classification data were used. With this analysis, buildings could be classified as being in areas without access to GSs (i.e., no GSs within the 300 m circular buffer) vs. with access to GSs (presence of GSs within the buffer). A sensitivity analysis performed in a previous study on the effect of vegetation on noise annoyance revealed that effects were similar for buffer sizes of 150, 300, 500 and 1000 m, and that a 300 m buffer yielded best explained variance [19]. Buildings that had more than one GS within the buffer were excluded from the analysis, in order to assign participants to a single GS to facilitate the interpretation of results.

The GSs were identified and selected in a stratified sample using land-use classification data of the Federal Swiss Office of Topography (swisstopo; same data as in [19]). GSs with restricted access (e.g., private/household gardens or playgrounds of schools) or with access requiring payment, namely sport fields (e.g., for soccer and golf), camping grounds, open-air swimming pools as well as the zoo of Zurich, were excluded. Initially, a total number of 124 GSs in the city of Zurich were included in the dataset. These GSs were divided into large (≥10,000 m^2^) and small (<10,000 m^2^) and subsequently, into loud and quiet (see details in Section 2.3). Twenty-three loud and 25 quiet GSs were identified; the remaining 76 that matched neither of the two groups were excluded from the study. The final dataset with the four groups of GSs included: (i) loud and large (*n* = 11), (ii) loud and small (*n* = 12), (iii) quiet and large (*n* = 18) and (iv) quiet and small (*n* = 7).

The vegetation-around-home (VEG-H), i.e., the residential green or greenness within buffers with a radius of 50 m around home locations, was also derived as a proxy for both access to green on the property and view from home on outdoor vegetation. The latter was found to be a crucial parameter for alleviating noise-induced health effects (see, e.g., [19,20]). To quantify VEG-H, the satellite-based indicator for greenness, Normalized Difference Vegetation Index (NDVI) [21], was used. Mean NDVI values for the months of April to October in the years 2019 to 2021 were used (data extracted from ESA [22]).

Based on the combinations of different levels of noise exposure and access to GSs, the design included the following study groups: one group with low noise exposure at home with access to quiet and large GSs (LA), one with low noise exposure at home but no GS access (LNA), four groups with high noise exposure at home with access to GSs (specifically to quiet and large (QuLa), quiet and small (QuSm), loud and large (LoLa) and loud and small (LoSm) GSs), and one with high noise exposure at home but no GS access (HNA). This provided seven study groups between which a variation in the stress levels (as measured by perceived and physiological stress) is expected (Figure 1).

### 2.3. Noise Exposure Assessment

Data from the Swiss noise database sonBASE for the year 2015 were used for determining the noise exposure classification of the study sites (both exposure at home and in the GSs) [23]. Data were available for road traffic and railway noise in a 10 m × 10 m resolution grid and 150 m × 150 m for aircraft noise. Daytime road traffic noise level *L*_day_, i.e., the yearly averaged A-weighted equivalent continuous sound pressure level over 16 h, from 6 a.m. to 10 p.m., was determined for the centroid coordinate point of each building.

Noise exposure at home was classified into either “*low road traffic noise exposure*” with *L*_day_ ≤ 53 dBA, or “*high road traffic noise exposure*” with *L*_day_ ≥ 68 dBA. Further, with the goal of not mixing different noise sources that may disparately impact health, houses exposed to railway noise of *L*_day_ > 54 dBA and aircraft noise *L*_day_ > 45 dBA were excluded from the study area. The lower threshold for road traffic and those for railway and aircraft noise were selected according to the recommendations of the WHO [24]. The higher threshold for road traffic noise (i.e., *L*_den_ of 68 dB) was derived from Brink et al. [25], which found 25% of the Swiss population to be highly annoyed at this level. Note that these thresholds are defined for the *L*_den_. As the *L*_den_ is not available from the sonBASE calculations, the *L*_day_ had to be used as an approximation. The two metrics are similar in situations with dominating daytime traffic, as is the case for the city of Zurich. In fact, Brink et al. [26] estimated a difference between *L*_den_ and *L*_day_ of approximately 2 dBA, which seems acceptable for the classification task at hand.

GSs were classified by road traffic noise exposure *L*_day_ (in addition to size). The *L*_day_ is appropriate for GSs since this is when visitors primarily spend time in public GSs. As road traffic noise levels may vary substantially within the area of larger green spaces, we used a GIS-based analysis for the GSs of Zurich and derived the following definitions for quiet and loud GSs:(1)If more than 50% of the GS area had *L*_day_ below 45 dBA, the GS was considered quiet.(2)If more than 50% of the GS area had *L*_day_ above 58 dBA, the GS was considered loud.

The lower limit was determined based on existing literature. The EU Working Groups on Assessment of Exposure to Noise and on Health and Socio-Economic Aspects proposed noise limits for quiet areas in cities, namely *L*_den_ 40 to 45 dB for garden and communal areas and a limit of 45 to 50 dBA for areas for outdoor activities [27]. The British Department for Environmental, Food and Rural Affairs (DEFRA) recommends a limit of 40 dB *L*_den_ as a “gold standard” for quiet areas in agglomerations. The same report shows that *L*_den_ of 45 to 50 dB is used as a criterion in some European countries [28]. Similarly, a group from Sweden proposed a limit of 45 to 50 dBA *L*_Aeq_ for GSs [29]. To achieve a similarly large gap between quiet and loud for the noise exposure in GSs as at home, we set the higher limit to *L*_day_ of 58 dBA. An even higher limit would have excluded too many potential GSs.

At the time of stratification described above, *L*_day_ was used as a proxy for *L*_den_ due to availability. For the main analysis, however, *L*_den_ will be available from calculations by the city of Zurich. These will yield exposure data for façade points of all dwellings (highest and lowest exposure per dwelling, based on façade points), expressed as *L*_den_ (primary variable for noise annoyance and stress [19,25]) and *L*_night_*,* (primary variable for sleep disturbance).

Exposure data will also be obtained from the field measurements to capture the (psycho-)acoustic metrics (cf. Section 2.4). Rather than being used as primary acoustical predictor metrics, these may be used to refine models.

### 2.4. Field Measurements in GSs

Audio recordings, as well as 360° images, were taken in the GSs to: characterize the noise exposure in more detail (classic acoustical and psychoacoustic parameters), identify audible noise sources other than road traffic and quantify greenness by counting green pixels in visual analysis. The measurement devices are listed in Table 1.

Field measurements were carried out in areas where visitors were likely to spend their time (i.e., close to benches, playgrounds, fountains or walking paths). The measurement devices were set five to ten meters away from these places to not bother visitors and to avoid disturbances in the recordings. One to six measurement locations were selected per GS (i.e., one measurement location in small and up to six in large GSs). Several metrics were determined from the measurements, including *L*_Aeq_10′_ (short-term A-weighted equivalent continuous sound pressure level, calculated as energetic mean from two 5 min measurement intervals) and the psychoacoustic parameters loudness, roughness and sharpness [30]. The latter were determined with the software ArtemiS (version 9.0). Loudness is expressed here as *N5*__10′_, which represents the (cumulative) 5% percentile for both five minutes measurement intervals.

Two recordings were taken at each of the 78 measurement locations during the morning between 08:50 a.m. and 11:15 a.m. (2:25 h range) and the afternoon between 15:50 p.m. and 18:10 p.m. (2:20 h range), in all GSs to capture both low and high volumes of road traffic (Figure 2). The recordings were five minutes long each. Two recordings were done to capture a range of noise exposure for each location throughout the day. (Psycho-)acoustic metrics for every measurement location were obtained from both recordings, morning and afternoon. The majority of the recordings were done in September and October 2021 (recordings at only four measurement locations were carried out in Spring 2022). The temporal gap was chosen to avoid recordings being taken in seasons without foliage.

### 2.5. Key Outcomes and Underlying Concepts

In this study, a number of outcomes will be investigated. These are briefly introduced below.

Stress: Psychological stress is a feeling of emotional strain and pressure, while physiological stress represents a raised activation level of the autonomic nervous system, achieved through autoregulatory neural and hormonal reactions [31]. During acute stress, cortisol levels rise and pulsatility is maintained [32]. Stress is evoked by a stressor, which may be defined as an unwanted event or situation [33]. Stress can be assessed as self-reported psychological stress and/or physiological stress, the latter assessing the neuroendocrine arousal that affects both the humoral (body fluid response) and metabolic state of the organism [6]. This study evaluates self-reported stress as well as physiological stress via hormones cortisol and cortisone.Noise annoyance: a psychological, concept describing the negative, subjective response of humans to noise [34,35]. ISO/TS15666 describes annoyance as “one person’s individual adverse reaction to noise”. It covers noise-related disturbance, emotional or attitudinal response and perceived control or coping capacity with the noise situation [36]. In addition to capturing the reaction to noise exposure, annoyance may also reflect susceptibility to stressors and noise sensitivity. Further, it may reflect the perceived stress due to noise [37] and thus plays an important role as a mediator for stress.Noise-induced sleep disturbance: a major effect of environmental noise is that it may alter the quality of sleep thus having an impact on health [38]. Sleep disturbance may be assessed as a physiological response, for example using polysomnography [39,40] or as a self-reported outcome [41]. This study assesses self-reported sleep disturbance.

We also assess the following aspects that may affect stress and/or noise annoyance:Coping with stress: behavioral and cognitive efforts made by a person in order to manage demands that exceed personal resources [42].Locus of control: a personality trait that explains the degree to which one believes to have control over their actions based on outcomes of behavior [43].Noise sensitivity: a stable personal characteristic that affects one’s reactivity toward noise [44]. It thus varies the effects of noise depending on the individual and is commonly seen as a moderator for the effects of noise exposure [45].

### 2.6. Participant Selection, Recruitment and Survey Protocol

Inhabitants of the study sites are considered suited for participation if they lived for at least one year at the same location and are at least 18 years old. Data collection is performed in two to three waves, with an optional third wave in case of lower-than-expected participation. Potential participants are contacted by letter and invited to participate in an online field survey. They are also invited to contribute to a participatory geographical information system (PGIS) process to identify a point within their most visited GS (see “Phase I” below). In the second step, a subsample of participants are visited at home to collect hair cortisol and cortisone probes to analyze long-term physiological stress, take photos from their living room windows and ask questions about the perceived soundscape and view from home. For this subsample, additional exclusion criteria apply (see “Phase II” below). Furthermore, a non-response analysis (NRA) interview is carried out by telephone to address potential bias (see “NRA” below). The two-step approach, i.e., split into two different phases (Phase I–II), is performed as indicated in Figure 3. Study participation is voluntary.

The survey is conducted in German, in two to three waves (early and late summer of 2022; possibly with a third wave in the warm season of the following year). Addresses of potential participants are selected from the stratified sample, separately for each wave, from the official register data by the Residents’ Office of the city of Zurich. The office provides a maximum of 20% of the total participants’ addresses for each study group and wave.

Phase I—online questionnaire: Participants are contacted with an invitation letter that contains both a link and a QR code to access the online questionnaire (implemented in Maptionnaire; Mapita Oy, Helsinki, Finland). As a motivation to participate, a lottery among all participants is conducted. A reminder letter is sent four weeks after having sent the invitation letters to all persons that did not fill in the questionnaire.

Prior to starting the online questionnaire, each participant needs to confirm their consent for participation. The questionnaire covers six sections. These are (i) personal information (age, sex) and living situation (including whether participants have a garden at home or not), (ii) noise annoyance and sleep disturbance (assessed through the numerical ICBEN 11-point scale [36,46]), (iii) noise sensitivity, (iv) personality and health, including perceived stress, (v) employment situation and occupation and (vi) leisure time activities.

For the noise annoyance (and sleep disturbance) question, the numerical 11-point scale was used. It has the advantage of a more simple numerical scale compared to the verbal 5-point scale [36], and the equal spacing of the numerical scale allows the data to be treated as continuous in the analysis. The requirement to ask both scales instead of just one of these scales has recently been relaxed [35,36]. The binary variable *Highly Annoyed (HAnn)* is defined as 1 when one of the three top scale points on the 11-point scale is marked, i.e., 8, 9 or 10 (which corresponds to 28% of the scale length), and else as 0 [47]. Noise sensitivity is assessed with a 5-point numerical scale where 1 means “completely disagree” and 5 “fully agree” to the sentence “I am sensitive to noise” as well as with the 13-item NoiSeQ-R instrument [48]. Perceived stress is addressed by asking about the *ability to cope with stress* (a 5-point numerical scale where 1 means “I cope badly with stress” and 5 “I cope well with stress”), as well as *stress in private life* and *stress at work* (a 5-point numerical scale where 1 means “no stress at all” and 5 “very high stress”). Further behavior-related issues are assessed through the short scale for the assessment of locus of control [43]. To shed light on leisure time behavior and activities, a PGIS mapping component is used to identify the participant’s most visited GSs, which serves to contribute to a better understanding of the properties of the GSs [49]. Participants are asked to locate a point in their most visited GSs used for recreation. Objective landscape characteristics can then be obtained for this location. Information on the usage practices of GSs (even though not the main focus of this study), is collected via questions about their visits, including perception, motives, as well as frequency and duration over different seasons of the year. Participants are asked about their perception of the GS safety and maintenance as well as their motives for visiting (social cohesion, playing sport, enjoying nature, etc.). To restrict the length of the questionnaire, we did not ask about the perceived psychological and/or psychological effects of visits. These questions, however, are addressed within another work package of RESTORE.

Phase II—visits at home: In the second phase of the field survey, a subsample of participants is visited at home. Participants volunteering in this phase receive 100 Swiss francs as compensation at the end of the visit. Similar to Phase I, a signed consent form is required for participation. Data are collected in a paper-and-pencil questionnaire on: (a) health habits (as some habits might bias the physiological hair cortisol levels) (b) inclusion/exclusion criteria for participation, (c) the perception of the sound situation at home, and (d) the perception of the view from the participant’s living room window. To avoid potential bias, the following inclusion/exclusion criteria apply: Participants must be between 18 and 70 years old, neither be pregnant nor have given birth in the last 12 months (due to the lactation period and its influence on the cortisol levels), not suffer from morbid obesity (i.e., body mass index (BMI) > 35), not take cortisol-changing medication, not use chemical products for their hair (such as hair dying) and not have experienced a profoundly emotionally negative or positive event in the last 12 months (e.g., job loss, divorce, death of a beloved person, serious illness, marriage, having children). A photograph of the outside-view from the participant’s living room window is taken. The photo will be used for a viewshed analysis. By counting the pixels corresponding to the vegetation coverage, the greenness in each dwelling’s surroundings will be estimated. Lastly, a sample of the participant’s hair is taken to determine long-term stress. The hair samples will be segmented into 3 cm segments representing a time frame of the 3 months prior to collection. The samples will be analyzed at the Centre for Forensic Hair Analytics of the University of Zurich using liquid chromatography–tandem mass spectrometry (LC-MS/MS) [50]. A mean cortisol and cortisone level, representative of the last three months, will be available to quantify chronical physiological stress.

Non-response analysis: Since study participation is voluntary, the study might be susceptible to a non-response bias [51]. Therefore, a NRA will be conducted to assess the distribution of noise exposure to road traffic, noise annoyance, noise sensitivity, age, perception of GSs, perceived stress and educational level in non-responders as compared to responders A total of 10% of randomly selected non-responders will be contacted (80% of them by telephone and 20% through a social media platform with the purpose of reaching mainly young people who do not use a landline phone). For those approached by telephone, phone numbers are extracted from two publicly available Swiss websites (https://tel.search.ch and https://local.ch, accessed on 7 March 2021). A maximum of two calls to the telephone group will be attempted, whereas for those contacted through social media an informative message will be left, both aiming to get consent to make an appointment for a phone call.

Power analysis: A power analysis (approximated with an a priori analysis for noise annoyance with a small effect size using GPower version 3.1.9.7 [52]) suggested a target response rate of approximately 30%. This number corresponds to the response rate in a previous field survey on noise annoyance and sleep disturbance in Switzerland [25,53]. The Residents’ Office of the city of Zurich provided us with 5168 addresses from the targeted areas for waves 1 and 2. A total of 256 addresses were used to carry out the pilot study (Section 2.8). The remaining 4912 addresses will be used in waves 1 and 2 of the main field survey. (A third wave with additional participants will be conducted if the response rate is not reached.) For Phase II of the survey (i.e., the visit at home), a total of up to 300 hair cortisol and cortisone analyses are budgeted, assuming that ~20% of the participants in the field survey also participate in the home visits. This is in accordance with estimates from another power analysis assuming a medium effect size for the physiological stress analysis. The significance level was set at 0.05 in both cases.

### 2.7. Data Analysis and Modeling

To test the hypotheses presented in Section 2.1, different models will be established. The link between the binary variable HAnn with road traffic noise exposure at home (*L*_den_) and access to GSs and/or VEG-H will be explored with logistic regression analysis. The binary variable HAnn will be studied for comparability with previous field surveys (e.g., [24,47]). The association between perceived stress with road noise exposure (*L*_den_) at home and access to GSs will be evaluated through an ordinal or linear regression analysis, treating stress either as an ordinal (5 levels) or continuous variable, respectively. (Ordinal variables may be treated as continuous if they have five or more categories, e.g., [54].) The approach for physiological stress is similar to that for perceived stress, except it is a purely continuous variable. The models will be adjusted for age, sex, socio-economic status, BMI, physical activity, smoking status and employment situation [55]. Perceived stress will additionally be adjusted for potentially stress-related factors such as stress in private and work life.

To complement these analyses, models using the NDVI within the 300 m buffer as a general indicator for residential greenness instead of GSs and/or VEG-H will be explored, because [19] found NDVI to be a particularly strong predictor for (reduced) noise annoyance. Further, structural equation models (SEM) will be implemented to explore the role of mediation. SEM have already been successfully applied to study the effect of transportation noise on annoyance and health-related quality of life [56]. Details on the modeling approaches will be set during the actual analyses.

### 2.8. Pilot Study to Test the Feasibility of the Study Protocol

A pilot study, consisting of Phase I of the study only, was conducted to test the feasibility of the study protocol and get an estimate for possible response rates. The two extremes of the seven study groups were included, specifically (i) *High noise exposure at home with no access to GSs* (HNA) and (ii) *Low noise exposure at home with access to GSs* (LA). A subsample of 256 participants was selected from both groups, and an invitation letter was sent.

Six weeks after sending out the invitation letter, a subsample of non-respondents was contacted by phone to increase the sample size and/or find out the reason why they did not participate in the study. None of these participants were willing or able to subsequently fill in the survey. A reminder letter was also sent to another subsample from the original sample that neither had filled in the online questionnaire nor had been contacted via telephone yet.

## 3. Results

### 3.1. Implementation of the Study Concept

Figure 4 shows the 48 GSs within the city of Zurich, the areas assigned to the seven study group study sites and the 78 measurement locations (Section 2.4).

The city of Zurich had a population of 421,878 by the year 2020 [57]. A total of 107,546 persons were living in the study sites at the time the study design was implemented (February 2021), of whom 38,899 persons were eligible for participation (time of data delivery by the City of Zurich in February 2022) (inclusion/exclusion criteria in Section 2.6) (Table 2).

### 3.2. Field Measurements

Figure 5 shows the distribution of measured noise exposure (see Section 2.3) between the four types of GSs (loud-large, loud-small, quiet-large, quiet-small). The assignment of GSs based on noise maps was confirmed as GSs classified as loud were characterized by higher noise exposure than those classified as quiet. In addition, the small GSs tended to show higher noise exposure than the large ones, which can be explained by their greater proximity to traffic infrastructure.

Figure 6 shows the psychoacoustic loudness *N5*__10′_ and short-term *L*_Aeq_10′_ of the GSs assigned to five of the seven study groups (*HNA* and *LNA* do not have access to GSs and thus have nothing to measure). Both indicators discriminate the GSs assigned to the study groups well and in the expected order. The two groups *LA* and *QuLa* were those with the lowest *N5*__10′_ and *L*_Aeq_10′_ values. In addition, the study group *LoSm* had the highest *N5*__10′_ and *L*_Aeq_10′_. Further, the two indicators *N5*__10′_ and *L*_Aeq_10′_ were highly correlated (Spearman R = 0.96, *p* < 0.0001).

The relation between the *L*_day_ obtained from the sonBASE database and the *L*_Aeq_10′_ obtained from the measurements was assessed with a Pearson correlation analysis (Figure 7). Each data point represents a measurement location and is classified by the type of GS according to the noise level (loud or quiet). A moderate correlation was found for the loud GSs (*r* = 0.36), where the measurements yielded similar exposure values as calculations, which in turn indicated that road traffic is the dominant noise source. The quiet GSs, in contrast, were characterized by louder measured sound levels and depicted a weaker correlation (*r* = 0.19). This indicated that sources other than road traffic dominate the noise exposure. None of these two groups showed significant correlations (*p* > 0.05).

### 3.3. Pilot Study Insights

From the 256 persons contacted with the invitation letter, 26 participants completely filled in the online questionnaire. Of the 49 non-respondents successfully contacted by phone, 26 persons indicated their reason for non-participation. These included disinterest (69%), questionnaire forgotten (8%), lack of time (4%) and other (19%). The reminder letter was sent to 180 persons. Fifteen recipients of this reminder letter completed the survey, raising the number of participants to 41 (Figure 8).

A similar response rate was obtained in the pilot study through the reminder letter (8.3%) as through the invitation letter (10.1%), giving a total rate of valid responses of 16% (Table 3). Table 3 depicts the evolution of data collection process. The total sample drop out (i.e., the loss of participants who started but did not finish the questionnaire) is non-negligible (2.7%). The reasons for this are unknown and, consequently, cannot be prevented.

Of the 41 participants completing the questionnaire, 24 belonged to the study group LA (58%) and 17 to HNA (42%). Table 4 summarizes the living situations of these groups in terms of noise exposure and VEG-H. Overall, the HNA group was substantially louder (more than 15 dBA) than the LA group and considerably less green (24% lower NDVI value). However, both areas were relatively green, as NDVI values > 0.3 indicate higher and denser vegetation and/or forests [21].

Figure 9 shows the ratings of noise annoyance and sleep disturbance among participants living in the two study groups HNA and LA, separately for the three transportation noise sources road traffic (cars and trucks), railway and aircraft. While no high levels of railway and aircraft noise were expected given the site selection criteria (cf. Section 2.3), the corresponding outcomes were nevertheless inquired about in the questionnaire. Although no significant differences were found between the two groups (*p* > 0.05), the participants of the HNA group tended to be more annoyed by road traffic, while the opposite was observed for aircraft noise. While the first outcome was expected, the second was rather surprising as aircraft noise levels were kept low in all study areas. This finding might be supported by results from [19] who identified a substantially higher annoyance for inhabitants of green areas compared to people living in areas with less green at a similar aircraft noise exposure. The self-reported sleep disturbance confirmed the trends seen for noise annoyance. Annoyance and sleep disturbance were generally very low for railway noise, which was expected given the low exposure.

While differences between the HNA and LA groups were non-significant, there were significant differences between highly annoyed and non-highly annoyed persons regarding their exposure to *L*_day_ and VEG-H. In contrast, no significant differences were observed in the ability to cope with stress and stress in private life (Table 5).

## 4. Discussion

This paper presents a study protocol for a cross-sectional study with a large field survey (online questionnaire followed by visits at home) on the association of noise annoyance and long-term stress with road traffic noise exposure and vegetation greenness. The study concept uses a classification scheme for different residential areas regarding access to GSs and road traffic noise exposure at home, to obtain a stratified representative sample. The feasibility of the study protocol was successfully proven in a pilot study, which, despite the small sample size, revealed the first insight into trends on the association of noise annoyance and perceived stress with road traffic noise exposure and GSs.

As the sample size of the pilot study was quite small, the results are hardly representative of the population of Zurich. The pilot study also focused only on the two types of living situations with the most extreme exposure conditions to road traffic noise and GSs. Thus, these pilot results are constrained to these extreme conditions (LA and HNA) and should be interpreted cautiously.

Even though the COVID-19 pandemic was still present at the time of the field measurements in the GSs, outdoor activities were already at a regular level again in Switzerland. We thus consider the measurements as representative of the post-pandemic, “normal” situation.

In interpreting the correlation between the measured and calculated noise exposure levels, one should consider that the measurements were intended as a complement to the SonBASE database rather than a validation. In particular, our recordings lasted only a few minutes each, so a comparison with yearly averages needs to be interpreted with care. Nevertheless, some conclusions can be drawn from the comparison of the data. While the measurements yield similar noise levels as the calculations in the loud GSs, indicating that road traffic noise dominated these areas, they revealed substantially higher noise levels than the calculations in the quiet GSs likely due to the low road traffic volumes and dominance of other noise sources. Thus, measurements provide additional information that is not (necessarily) contained in the calculations. Moreover, from the measurements, psychoacoustic parameters such as loudness can be obtained. These may describe the soundscape of the GSs in greater detail than classical modeled noise metrics, and might therefore be used as secondary predictor variables for annoyance and stress. The correlation analysis further revealed that the scatter of the data was large for both measurements and calculations. This was attributed to the specific measurement locations, particularly within large and loud GSs, that were close to a road. As such, they yielded high (calculated and measured) noise exposure levels even though the GS, as a whole, had been classified as quiet.

The response rate obtained with the reminder letter in the pilot study was similar to the rate of the initial invitation letter, which suggested the reminder letter was crucial for a sufficiently high response rate. As the valid response rate of 16% was substantially lower than the aspired 30%, a third wave is very likely to be necessary for the main study. Whether the aspired response rate of 20% for the visits at home will be reached will only be revealed during the main study.

Based on the study protocol developed here, the main field survey is currently carried out. The results will allow exploring and quantitatively assessing, among others, the benefits of the GSs regarding noise annoyance and psychological and physiological stress. Regarding residential greenness, we parametrization access to public GSs through a binary variable (presence/absence), based on Euclidian proximity from the residents, as was previously done in epidemiological studies (e.g., [58,59]). Thus, we link access and proximity for our classification scheme of the study areas. The distance covered by the buffer size of 300 m selected here is assumed to be walkable within 6 min (given that a fairly direct route to the GS is available). This threshold has been proposed by international standards and policy objectives for local green space planning [60]. The link between access and proximity should thus be sufficiently close to justify our paratremization of access to GSs. In other words, the GS in the neighborhood could be easily reached regardless of the resident’s physical condition, and hence, residents could gain benefit from it as indicated in many epidemiological studies [61,62,63,64]. Since stress can also be reduced by spending time in one’s own household garden [65] or even via the view from home of green areas [20], we also consider the residential greenness immediately around participant’s dwellings. To do so, we use the index vegetation-around-home (VEG-H) described by the mean NDVI within a small buffer around residents’ homes, which, as a proxy for the view from home on outdoor vegetation, has been found in several studies to be important to reduce noise-induced effects (e.g., [19,20,66]).

## 5. Conclusions

Field measurements as well as the application of the study protocol in a pilot study revealed that the protocol was suitable to stratify the population of a large city at different levels of noise exposure as well as access to green spaces. Based on this, it can thus be applied in the current main study. Indeed, despite the small sample size, results of the pilot study indicated that noise annoyance and sleep disturbance may be higher among residents living in areas with less greenery and increased road traffic noise exposure at home compared to those in rather quiet and green residential areas. The classification scheme for different residential areas regarding access to green spaces and noise exposure is thereby a core element to obtain a representative stratified sample that might also be useful in other projects. Our study protocol and its application may complement previous research and bring new insights into the associations of noise annoyance, stress, noise exposure and green spaces. The main survey, based on the presented study protocol, is currently running. This work could thus contribute to updating the legal basis in densely populated urban areas.

## Figures and Tables

**Figure 1 ijerph-20-03203-f001:**
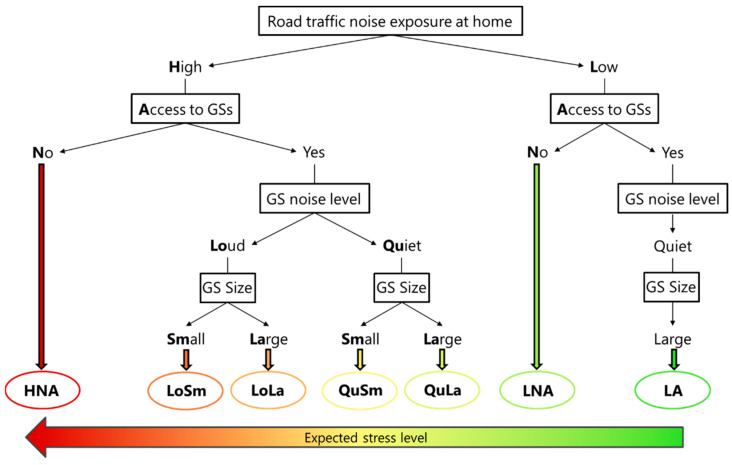
Classification of the seven study groups accounting for road traffic noise exposure at home in combination with access to GSs of varying sizes. The stress level is expected to increase from right to left.

**Figure 2 ijerph-20-03203-f002:**
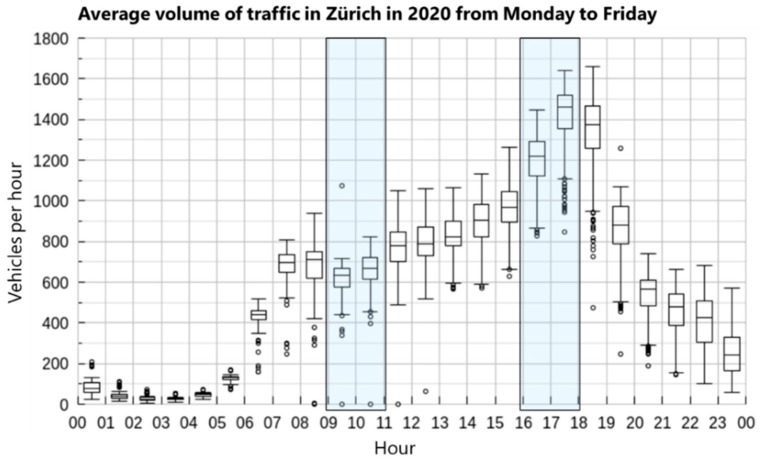
Average daily road traffic counts (vehicles per hour) at a measurement location in central Zurich in the year 2020 with the hours where the measurements took place highlighted. Source of data: City of Zurich (2021).

**Figure 3 ijerph-20-03203-f003:**
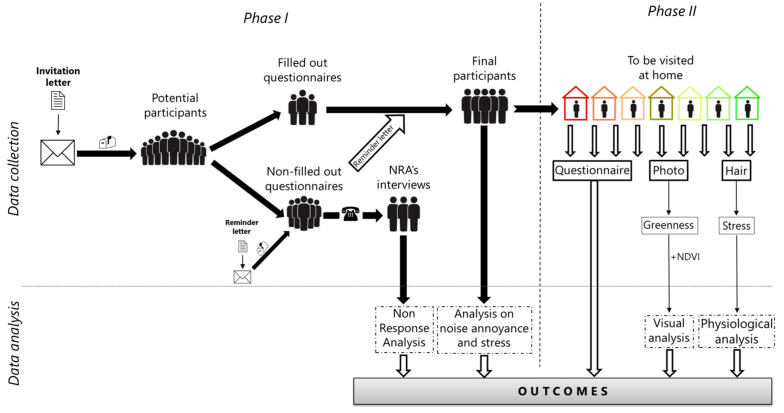
Graphical overview of the workflow chart of the field survey.

**Figure 4 ijerph-20-03203-f004:**
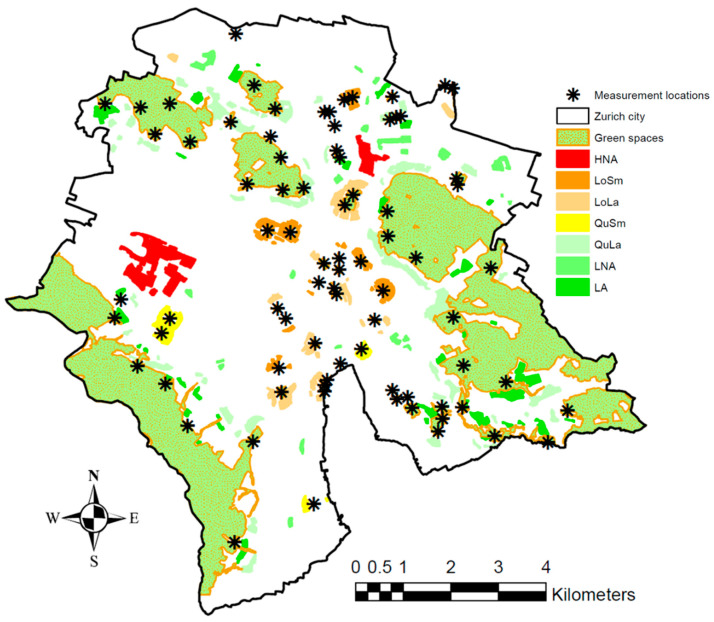
Areas assigned to the seven study groups: *High noise exposure at home with no access to GSs* (HNA), High noise exposure at home with access to small and loud GSs (LoSm), High noise exposure at home with access to large and loud GSs (LoLa), *High noise exposure at home with access to small and quiet GSs* (QuSm), *High noise exposure at home with access to large and quiet GSs* (QuLa), *Low noise exposure at home with no access to GSs* (LNA) and *Low noise exposure at home with access to large and quiet GSs* (LA) (cf. Figure 1). As additional information, the 48 GSs included in the study and the 78 measurement locations in these GSs (see Section 2.4) are shown.

**Figure 5 ijerph-20-03203-f005:**
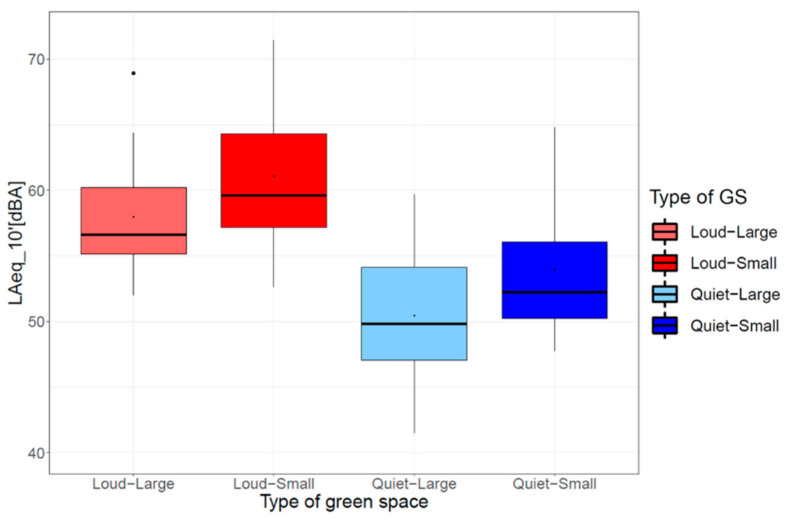
Boxplots with mean (point in boxes), median (horizontal line in boxes), the first and third quantiles (25% and 75%, lower and upper boundaries of boxes), whiskers comprising the data within 1.5 times the interquartile range and outliers outside the whiskers of the short-term A-weighted equivalent continuous sound pressure level, *L*_Aeq_10′_ in GSs classified by size and noise exposure.

**Figure 6 ijerph-20-03203-f006:**
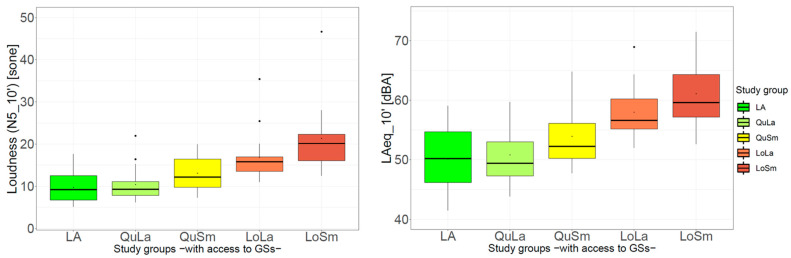
Boxplots of the psychoacoustic and acoustic parameters among different study groups: (**left**) loudness-*N5___*_10′_; (**right**) *L*_Aeq_10′_. Study groups “Low noise exposure at home with No Access to GSs (LNA)” and “High noise exposure at home with No Access to GSs (HNA)” are not shown as these have no access to any green spaces. Note the different scales of loudness and *L*_Aeq_10′_.

**Figure 7 ijerph-20-03203-f007:**
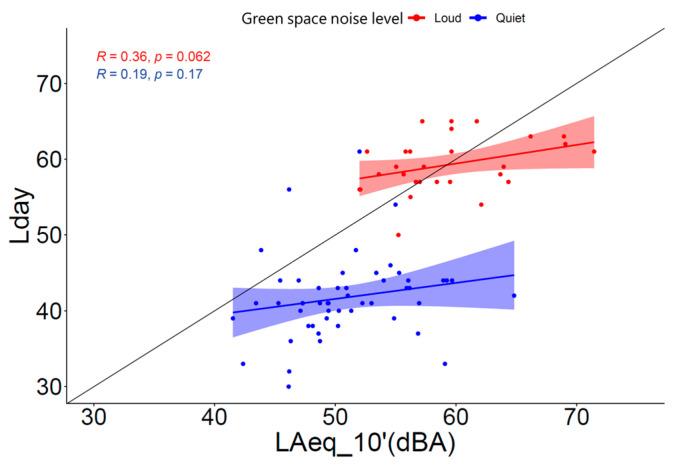
Correlation between *L*_day_ obtained from the sonBASE database and *L*_Aeq_10′_ obtained from measurements with Pearsons’ correlation coefficients (*R*) and *p*-values.

**Figure 8 ijerph-20-03203-f008:**
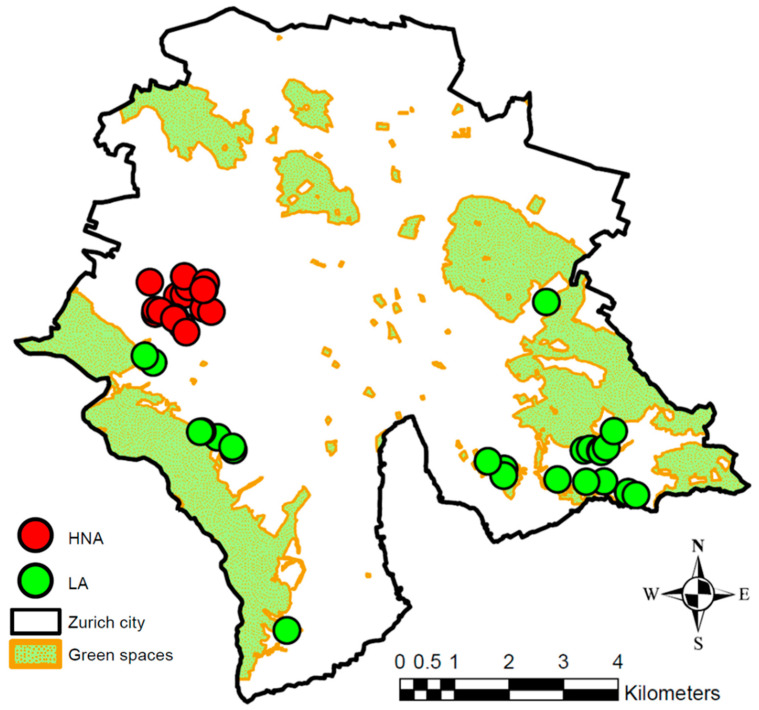
Pilot study locations of the participants’ residences buildings in subgroups HNA (*High noise exposure at home with No access to GSs*, in red) and LA (*Low noise exposure at home with access to GSs*”, in green) and GSs included in the study. The participants’ locations are approximate for data privacy purposes.

**Figure 9 ijerph-20-03203-f009:**
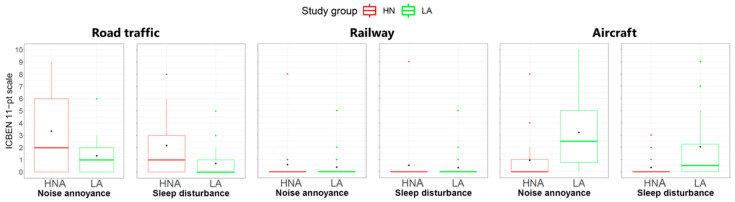
Boxplots of noise annoyance and self-reported sleep disturbance for road traffic (**left**), railway (**center**) and aircraft noise (**right**) for the two study groups *High noise exposure at home with No access to GSs* (HNA, in red) *Low noise exposure at home with access to GSs* (LA, in green).

**Table 1 ijerph-20-03203-t001:** Devices used in the study for field measurements.

Device	Manufacturer	Measurement
NTi XL2	NTi Audio	Audio recordings
QooCam 360 camera	Kandao	360° images
Hydromer A1	Rotronic	Temperature
Stratos 3 A1929	Amazfit	Atmospheric pressure and geographic coordinates
Windmaster 2	Kaindl	Wind speed

**Table 2 ijerph-20-03203-t002:** Numbers (*n*) and shares (%) of persons in total eligible for participation in each of the study areas compared with the city of Zurich. The percentages refer to the total population of the city of Zurich.

	No. Persons		Eligible Persons	
	*n*	%	*n*	%
City of Zurich	421,878	100	-	-
LA	12,211	2.9	4874	1.2
LNA	9594	2.3	3667	0.9
QuLa	33,867	8.0	8974	2.1
QuSm	6010	1.4	2234	0.5
LoLa	12,882	3.1	4420	1.0
LoSm	15,940	3.8	6288	1.5
HNA	17,042	4.0	8442	2.0
Total	107,546	25.5	38,899	9.2

**Table 3 ijerph-20-03203-t003:** Numbers (*n*) and rates (%) of the sample obtained at different stages of the pilot study.

	Invitation Letter		Reminder Letter		Total	
	*n*	%	*n*	%	*n*	%
Invited	256	100	180	100	256	100
Respondents	28	10.9	20	11.1	48	18.8
Drop out	2	0.8	5	2.8	7	2.7
Valid respondents	26	10.1	15	8.3	41	16

**Table 4 ijerph-20-03203-t004:** Road traffic noise exposure (*L*_day_) and VEG-H (NDVI) at home of the participants (50 m buffer) of the groups *Low noise exposure at home with Access to GSs* (LA) and *High noise exposure at home with No access to GSs* (HNA).

Study Group	Road Traffic *L_day_* (dBA)			VEG-H (NDVI)		
	Mean	Median	Sd	Mean	Median	Sd
LA	40.9	04.7	2.41	0.65	0.64	0.06
HNA	56.2	55.8	1.23	0.41	0.40	0.11

**Table 5 ijerph-20-03203-t005:** Differences in stress and exposure characteristics in a 50 m buffer around home between the non-highly (Non-HAnn) and highly annoyed persons (HAnn) by road traffic noise, and *p*-values.

		Non-HAnn	HAnn	*p*
Ability to cope with stress	Mean (SD)	3.0 (1.0)	3.8 (1.0)	0.181
Stress in private life	Mean (SD)	2.6 (1.1)	2.8 (2.1)	0.842
Road traffic *L*_day_ (dBA)	Mean (SD)	46.2 (7.5)	57.5 (0.7)	0.005
VEG-H (NDVI)	Mean (SD)	0.6 (0.1)	0.4 (0.0)	0.018

## Data Availability

The data presented in this study are available in the Appendix A.

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
