# Peer review of "How Do Road Traffic Noise and Residential Greenness Correlate with Noise Annoyance and Long-Term Stress? Protocol and Pilot Study for a Large Field Survey with a Cross-Sectional Design"

_ijerph, 2023, doi:10.3390/ijerph20043203_

Round 1

Reviewer 1 Report (Previous Reviewer 1)

The authors have addressed my comments and recommendations on the initial version with much care and in a very adequate way.

Author Response

We thank Reviewer 1 for his positive feedback.

Reviewer 2 Report (Previous Reviewer 2)

The issues are addressed according to the comments. The manuscript can be accepted after rechecking the grammar and typos.

Author Response

We thank Reviewer 2 for his/her feedback.

A native speaker has checked the manuscript and has introduced several language improvements (see new version).

Reviewer 3 Report (New Reviewer)

The discussion must be more extended and the conclusions can be more direct 

Author Response

We thank Reviewer 3 for his/her feedback.

We have consolidated the conclusion. However, for the discussion, we only introduced minor changes as we did not find any additional points that seemed important enough to discuss.

This manuscript is a resubmission of an earlier submission. The following is a list of the peer review reports and author responses from that submission.

Round 1

Reviewer 2 Report

This paper aims to assess the effects of greenness spaces on the public noise annoyance and the related unhealthy situations. The authors have a detailed description on the methodologies. However, due to the limited samples received from the selected participants, no solid conclusions can be drawn. Besides, for the methodologies, there are confusing parts that are not well discussed. For example, in lines 325-330, the definitions of quite and loud GSs were proposed. Is there any underlying basis for such classification? For the field measurements, it was stated in lines 361-362 that the measurement locations for the small GSs and large GSs were at least one or six, respectively. How do you determine all these locations and how do you process the data obtained from these field measurements? In general, the results presented were over the entire Zurich city, while the sample size seems not big enough. I will suggest to have a study that can concentrated on a typical so-called small or large GSs with different noise levels, by which more substantial results can be obtained. Additional specific comments are as follow.

1. (1) Please check the written errors in the manuscript. For example, the error in the line 284 (.05) and line 448 (.001). (2) Please check the units of the acoustic metrics in the pictures (e.g. Figure 6) and in the text, the unit of Lday should be dBA instead of dB.

2. In line 327~330, how to determine whether the 50% GS is below 45 dBA or above 58 dBA?

3. In line 361~362, Could the authors explain why “At least one measurement location was selected in all GSs and up to six in the case of large ones.”? How many monitoring sites did the authors actually use in the field experiment? And were all the measurement locations in Figure 2 used to record sound? It' s confusing.

4. In Figure 8, what is the purpose of the validation of the sonBASE results and the field measurements? The results of the figure 8 seems not good.

5. Since the sonBASE is available, the authors may consider combining the validated noise maps (generated by sonBASE) with the pilot study to provide more analysis for this manuscript. And the validated noise maps can replace the large-scale field measurements,which can avoid unnecessary workload.

Reviewer 3 Report

1.       In the introduction, the characteristics of road traffic noise should be added. There are many sources of urban noise. Why do you choose road traffic noise as the research object? What is the difference between it and other noises?

2.       Compared with the detailed data and methods, the results seem unappealing. Try to supplement the analysis with the concentration of the data distribution.

3.       The axes of figure 7 and figure 9 show not very clear, suggest the author to replace the pictures from the new.

4.       The conclusion of the paper is not novel enough. It is recommended that the author strengthen the conclusion.

5.       In this paper, what are the main and interesting findings?